# Elucidation of Mechanical, Physical, Chemical and Thermal Properties of Microbial Composite Films by Integrating Sodium Alginate with *Bacillus subtilis* sp.

**DOI:** 10.3390/polym13132103

**Published:** 2021-06-26

**Authors:** Charles Ng Wai Chun, Husnul Azan Tajarudin, Norli Ismail, Baharin Azahari, Muaz Mohd Zaini Makhtar

**Affiliations:** 1School of Industrial Technology, Bioprocess Technology Division, Universiti Sains Malaysia, Penang 11800, Malaysia; charlesngwaichun@student.usm.my (C.N.W.C.); muazzaini@usm.my (M.M.Z.M.); 2School of Industrial Technology, Environmental Division, Universiti Sains Malaysia, Penang 11800, Malaysia; 3School of Industrial Technology, Bioresource, Paper and Coatings Division, Universiti Sains Malaysia, Penang11800, Malaysia; baharin@usm.my

**Keywords:** materials, microbial composite films, sodium alginate, *Bacillus Subtilis*, reinforcing materials

## Abstract

Materials are the foundation in human development for improving human standards of life. This research aimed to develop microbial composite films by integrating sodium alginate with *Bacillus subtilis*. Sodium alginate film was fabricated as control. The microbial composite films were fabricated by integrating 0.1, 0.2, 0.3, 0.4, 0.5 and 0.6 g of *Bacillus subtilis* into the sodium alginate. Evaluations were performed on the mechanical, physical, chemical and thermal properties of the films. It was found that films reinforced with *Bacillus subtilis* significantly improved all the mentioned properties. Results show that 0.5 g microbial composite films had the highest tensile strength, breaking strain and toughness, which were 0.858 MPa, 87.406% and 0.045 MJ/m^3^, respectively. The thickness of the film was 1.057 mm. White light opacity, black light opacity and brightness values were 13.65%, 40.55% and 8.19%, respectively. It also had the highest conductivity, which was 37 mV, while its water absorption ability was 300.93%. Furthermore, it had a higher melting point of 218.94 °C and higher decomposition temperature of 252.69 °C. SEM also showed that it had filled cross-sectional structure and smoother surface compared to the sodium alginate film. Additionally, FTIR showed that 0.5 g microbial composite films possessed more functional groups at 800 and 662 cm^−1^ wavenumbers that referred to C–C, C–OH, C–H ring and side group vibrations and C–OH out-of-plane bending, respectively, which contributed to the stronger bonds in the microbial composite film. Initial conclusions depict the potential of *Bacillus subtilis* to be used as reinforcing material in the development of microbial composite films, which also have the prospect to be used in electronic applications. This is due to the conductivity of the films increasing as *Bacillus subtilis* cell mass increases.

## 1. Introduction 

Materials are the foundation for providing milestones in human development and improving human standards of life and production [1]. New materials are the foundation for emerging technology. Due to the rapid development of modern science and technology which focuses on industrial growth, economy and environmental protection, there are now more stringent and precise specifications for materials. Nowadays, material science is progressing towards the form of materials constructed according to specified properties. For this purpose, high-performance composite materials were developed to substitute or strengthen most of the other materials that existed in the 20th century [1]. The development and evolution of such composite materials for the past few decades is a class example of material design in human history.

A composite material is a type of complex multiphase multicomponent system consisting of a matrix material and a reinforcing material and composed of a matrix phase, a reinforcement phase and an interphase [2]. Materials Comprehensive Dictionary’ by Fazeli et al. (2019) provides a more specific and detailed description of composite materials: “Composite materials are new materials that are combinations of different types of materials, such as organic polymers, inorganic non-metal or metal, etc.” [3]. Materials design may allow the performance of each component to balance each other and interrelate with each other, culminating in new performance dominance that has critical variations from mixed general materials. It not only preserves the core function of the original product materials but also provides outputs that are not represented by the integrated effects of the original components [4]. Composite materials are usually made in industry through the combination of two materials. One of the materials is the matrix or binder, and another material is the reinforcement filler. The two materials often have very different characteristics but may work together to create a composite with unique characteristics. The matrix and filler can be easily distinguished within the composite, since they do not dissolve or mix into each other. The composite material fabricated in this research is called the microbial composite film, where the matrix material employed was sodium alginate, while the reinforcing material was *Bacillus subtilis*. Therefore, the intention of this research was to incorporate sodium alginate films with various amounts of *Bacillus subtilis* for the development of microbial composite films.

Some of the previous research that is related to microbial composite films includes the production of a multilayer conductive bacterial-composite film by embedding an electroactive bacterium, *Shewanella oneidensis* MR-1, within a conductive three-dimensional poly(3,4-ethylenedioxythiophene)/poly(styrenesulfonate) (PEDOT/PSS) matrix electropolymerized on a carbon felt substrate (MCBF) [5]. There are also composite films that are fabricated using bacterial cellulose, such as films that are incorporated with bulk chitosan and chitosan nanoparticles. This kind of film is aimed at providing new biodegradable food packaging [6]. Moreover, antimicrobial edible film for food packaging made from bacterial cellulose nanofibers, starch or chitosan is also possible, as studied by Abral et al. [7]. 

Bacteria, well-known for their small structure, typically a few micrometers in length, were among the first forms of life and the simplest organisms evolved on Earth and are present in every corner of the world. Bacteria are the most abundant and omnipresent life form on Earth that plays a vital role in both productivity and the cycling of substances that are essential to all other life forms [8]. They are also very useful in many applications, such as in the production of animal feed, useful chemicals, iron removal and electricity generation [9,10,11,12]. *Bacillus subtilis* is a Gram-positive, rod-shaped bacterium found in soil and the gastrointestinal tract of ruminants and humans. It is considered a benign organism, as it is not pathogenic or toxigenic and does not possess traits that cause disease in humans, animals or plants. The potential risk associated with the use of this bacterium in fermentation facilities is low. *Bacillus subtilis* is also proven to have a significant effect on the self-healing of cracks in concretes [13]. Therefore, it was used in this research project to study its effects on the development of microbial films. Additionally, *Bacillus subtilis* is a potential conductive agent and has great potential in furthering the research of microbial composite films through fabrication of conductive biowire by introducing metal ions into *Bacillus subtilis.* Having a fully biodegradable and environmentally friendly biowire would give a huge advantage and space for innovation to the current electronics industry.

This research mainly focused on the utilization of *Bacillus subtilis* as the filler in developing microbial composite films. Investigations were carried out to study the effect of different cell mass (gram) of *Bacillus subtilis* in the development of microbial composite films. *Bacillus subtilis* cell masses used in this research were harvested during the log phase of cultivation. Figure 1 shows a freeze-dried form of *Bacillus subtilis*. Figure 2 shows a microbial composite film with the dimensions of 16 cm (length) × 16 cm (width).

## 2. Materials and Methods

*Bacillus subtilis* was obtained from Laboratory of Bioprocess Technology Division, School of Industrial Technology, Universiti Sains Malaysia, Penang, cultured in Penassay broth and harvested after 22 h. The ingredients of Penassay broth are peptone, yeast extract, beef extract, sodium chloride, glucose, dibasic potassium phosphate and monobasic potassium phosphate. The cell mass was then freeze dried as shown in Figure 1 for the fabrication of microbial composite films. Microbial composite films were fabricated using a 16 cm (length) × 16 cm (width) × 1.5 cm (height) glass mold, and the product is shown in Figure 2. The film-forming solution was prepared by slowly adding 4 g of sodium alginate powder into 300 mL of distilled water [14]. Different amounts of *Bacillus subtilis*, 0.1, 0.2, 0.3, 0.4, 0.5 and 0.6 g, were added to six different solutions. The films were then transferred to dry in a drying oven at 40 °C for 24 h. Crosslinking of dried microbial composite films was performed by using a 2% calcium chloride (CaCl_2_) solution with the immersion method for 2 min. Then, the treated films were placed between blotting papers to prevent the curling of films during the drying process at ambient conditions. 

The microbial composite films were analyzed and tested for physical, mechanical and chemical properties. For mechanical analysis, the microbial composite films underwent analyses of tensile strength, breaking strain and toughness. Physical analysis included thickness, opacity and brightness. Then, the films underwent testing of conductivity and water absorption before further analysis of differential scanning calorimetry (DSC), thermogravimetric analysis (TGA), scanning electron microscopy (SEM) and energy-dispersive X-ray spectroscopy (EDX). The chemical analysis included in this research project was Fourier transform infrared spectrometry (FTIR). 

The tensile strength, breaking strain and toughness of microbial composite films were tested by using a Texture Analyzer based on the ASTM D882 standard test method. The thickness of the microbial composite films was determined based on the TAPPI T411 Standard (Abdul Khalil et al., 2017). The opacity of the microbial composite films was determined based on ASTM D1746-97, which is the standard test method for transparency of plastic sheeting. The brightness of the microbial composite films was determined based on ASTM D985, which is the standard test method for brightness of pulp, paper and paperboard. The conductivity of microbial composite films was determined based on ASTM F1711, which is the standard practice for measuring conductivity of thin films using the four-point probe method. Water absorption of the microbial composite films was determined according to the Water Absorption ASTM D570 method. Differential scanning calorimetry (DSC) of the microbial composite films was conducted based on the ISO 11357-1 standard method using DSC Q200 (TA Instruments, New Castle, DE, USA). Thermogravimetric analysis (TGA) of the microbial composite films was done based on the ISO 11358 standard method using TGA/DSC 1 (Mettler Toledo, Columbus, OH, USA). Scanning electron microscopy (SEM) and energy-dispersive X-ray spectroscopy (EDX) of microbial composited films were performed using FEI Quanta FEG 650 (Thermo Scientific, Waltham, MA, USA). Fourier transform infrared spectrometry (FTIR) of the microbial composite films was done based on the attenuated total reflectance-Fourier transform infrared spectroscopy (ATR-FTIR) method using IRPrestige-21 (Shimadzu, Kyoto, Japan). 

## 3. Results and Discussion

### 3.1. Analysis of Microbial Composite Films

This research aimed to integrate *Bacillus subtilis* and sodium alginate into a microbial composite film for reinforcement and strengthening. Various analyses on mechanical, physical, chemical and thermal properties were carried out to test the quality of the films and investigate what mass of *Bacillus subtilis* has the best effect on the reinforcement of microbial composite films. Material testing provides measurements of the characteristics and behavior of a substance. In this study, mechanical and physical testing were performed to provide information on the strength and physical properties of the films. Chemical analysis checked the composition of the films, while thermal analysis evaluated the properties of films as they changed with temperature. These analyses were included to characterize the films and provide more information and data for the study. 

### 3.2. Mechanical Testing 

Mechanical testing is performed to determine the specific mechanical properties of a material. The tensile strength, breaking strain and toughness of microbial composite films are tested on films and interrelated to each other. The tensile strength, breaking strain and toughness for sodium alginate films are 0.611 MPa, 84.372% and 0.016 MJ/m^3^, respectively. Sodium alginate films without adding any *Bacillus subtilis* acted as control in this research so that comparisons could be made between films with and without the reinforcement by *Bacillus subtilis.* In microbial composite films prepared by using *Bacillus subtilis,* the tensile strength increased from 0.611 to 0.858 MPa as the used *Bacillus subtilis* mass increased from 0.1 to 0.5 g. Then, the tensile strength dropped to 0.831 MPa in 0.6 g microbial composite films. The trend in breaking strain was the same as in tensile strength. It started with 84.849% in 0.1 g microbial composite films and increased to 87.406% in 0.5 g microbial composite films. In 0.6 g microbial composite films, breaking strain decreased to 86.155%. The toughness of microbial composite films increased steadily from 0.016 MJ/m^3^ in 0.1 g microbial composite films to 0.045 MJ/m3 in 0.5 g microbial composite films and then decreased to 0.031 MJ/m^3^ in 0.6 g microbial composite films. The films with 0.5 g of *Bacillus subtilis* had the best tensile strength, breaking strain and toughness compared to those of other films with different amounts of *Bacillus subtilis*. This indicated that sodium alginate films reinforced with 0.5 g of *Bacillus subtilis* could withstand greater stress and at the same time were tougher than sodium alginate films. This could be due to the fact that bacterial cells act as an effective reinforcing material in microbial composite films. Its submicron size provided a larger surface area for the interaction between bacterial cells and sodium alginate, thus providing a stronger bond and leading to a stronger material [4]. The reason why the value dropped after 0.5 g in the 0.6 g microbial composite film could be that the amount of 0.6 g was the maximum capacity of reinforcing materials that the films could contain and hence, a decrement was seen. Figure 3 shows graph of (a) tensile strength (MPa), (b) breaking strain (%) and (c) toughness (MJ/m^3^) of microbial composite films. Table 1 (Appendix A) shows the tensile strength (MPa), breaking strain (%) and toughness (MJ/m^3^) of the sodium alginate film and microbial composite films. 

### 3.3. Physical Testing 

The physical testing involved in this research comprised thickness, opacity, brightness, water absorption tests and scanning electron microscopy (SEM). The thickness of sodium alginate film without *Bacillus subtilis* was 0.578 mm. The thickness of the microbial composite films increased steadily from 0.665 mm in 0.1 g microbial composite films to 1.059 mm in 0.6 g microbial composite films. This was due to the fact that as the amount of *Bacillus subtilis* increased during the fabrication of microbial composite films, the thickness of the films increased. Additionally, the glass molds used to fabricate the films were of fixed dimensions, which were 16 cm (length) × 16 cm (width) × 1.5 cm (height) and thus, increasing the amount of bacterial cell mass contributed to the increase in thickness of the microbial composite films. However, thickness of films does not necessarily contribute to the strength and quality of the films and for that reason, in this study, 0.5 g microbial composite films had better quality in terms of tensile strength, breaking strain and toughness compared to 0.6 g microbial composite films. 

The opacity of the microbial composite films was tested using two different conditions, white light and black light. The opacity using white light and black light of sodium alginate films as control were 7.52% and 38.41%, respectively. In microbial composite films, white light opacity increased steadily from 8.1% in 0.1 g microbial composite films to 16.5% in 0.6 g microbial composite films. It can be seen from the results that white light opacity of microbial composite films prepared from all three growth phases had an increasing trend when the amount of bacterial cell mass increased. In the other case, black light opacity of films increased from 37.86% in 0.1 g films to 41.18% in 0.6 g films with slight fluctuations. Opacity provides an indication of how much light passes through a film [15]. The increase in the opacity of the films was due to the increasing bacterial cell mass that blocked the amount of light that passed through the films. 

The brightness of microbial composite films can be correlated to the opacity of films. The brightness of sodium alginate film as a control was 4.24%. In microbial composite films, the brightness of the films increased constantly from 6.65%in 0.1 g microbial composite films to 9.3% in 0.6 g microbial composite films. It can be concluded from the data that as the mass of bacterial cells increased during the fabrication of microbial composite films, the brightness of the films increased. Brightness describes how brilliant a sheet of paper appears. Brightness is a traditional measure that still appears on most packaging in the United States. The TAPPI standard (GE brightness) measures the ability of paper to reflect blue light. Whiteness measures paper in the same way the eye sees it. Light is made of all the colors combined. Paper brightness is measured on a scale of 0 to 100. This scale determines how much light is reflected from the surface of a sheet of paper. The higher the number, the brighter the paper. For example, paper with 98 brightness is slightly brighter than paper with 97 brightness. Therefore, it can be deduced from the findings that as the opacity of the microbial composite films increased, the brightness of the films increased as well. 

The conductivity of sodium alginate film as a control was 11.33 mV. In microbial composite films, the conductivity of the films increased constantly from 12 mV in 0.1 g microbial composite films to 37 mV in 0.5 g microbial composite films and then it dropped to 26.33 mV in the 0.6 g film. This increase in conductivity of the microbial composite films could be linked to the presence of *Bacillus subtilis* in the films. *Bacillus subtilis* is proven to be electrochemically active, and the electron transfer mechanism is mainly due to the excreted redox compounds (mediators) in the broth solution [16]. Redox mediators are compounds that speed up reaction rate by shuttling electrons from biological oxidation of primary electron donors or from bulk electron donors to electron-accepting organic compounds [17]. The binding of bacterial cells with the sodium alginate films created better linkages in-between and hence a better material for conductivity. Absorption of minerals present in the broth by *Bacillus subtilis* also aided in the increase of conductivity, as minerals are usually charged. Microbial composite film reinforced with 0.5 g *Bacillus subtilis* bacterial cell mass had the highest conductivity. This could be related to the fact that 0.5 g bacterial cell mass provided better reinforcing effects in terms of tensile strength, breaking strain and toughness. Better physical properties could possibly render better conditions for conductivity. 

The water absorption of sodium alginate film as a control was 264.29%. Regarding microbial composite films, the water absorption of the films increased constantly from 269.05% in 0.1 g microbial composite films to 307.78% in 0.6 g microbial composite films. This was mainly due to the absorption of water molecules by bacteria cells when the films were immersed in distilled water [18]. Both the charged and polar hydrophilic amino acid side chains in protein molecules can attract water molecules due to hydrogen bond formation. When water molecules were attracted to the outer surface of bacteria cells, osmosis occurred, as there was a different concentration gradient of water molecules between the outer and inner side of bacteria cells [19]. Hence, water molecules could enter the dry bacteria cells with low water molecule concentration, and this allowed more water molecules to be trapped in bacteria cells and absorbed as a part of the microbial composite film. However, the film should have a certain degree of water-resistant properties to prevent excessive water absorption from the surroundings, which can weaken the structure of film. Figure 4 shows the graph of (a) thickness (mm), (b) opacity (white light, %), (c) opacity (black light, %), (d) brightness (%), (e) conductivity and (f) water absorption (%) of the films. Table 2 (in the Appendix A) shows the thickness (mm), opacity (white light and black light, %), brightness (%), water absorption (%) and scanning electron microscopy (SEM) of sodium alginate film and microbial composite films.

### 3.4. Scanning Electron Microscopy (SEM)

The cross-sectional images of sodium alginate film and the surface of 0.5 g microbial composite films were examined with 5000× and 10,000× magnification, respectively, in order to provide high-resolution pictures of the surfaces of the two subjects. Referring to Figures 6 and 7, it could be seen that the cross-sectional part of sodium alginate film had obvious cleavages, while in the microbial composite film, the cleavages and voids were filled up significantly. The film without bacteria reinforcement also showed a rough and uneven surface with visible cracking along the surface, while the microbial composite film showed a smoother surface. This was mainly due to the cracking that occurred on the film itself. However, after the film was reinforced with *Bacillus subtilis*, it became more compacted with a slightly striated cross-section without cracking along the surface, typical of a stronger film with a more homogenous structure. This shows that *Bacillus subtilis* was able to fill into the voids between the components and enhanced the bonding of components in the film. Therefore, it contributed to a composite film which possessed improved mechanical properties. It is also clear that the surface of microbial composite films had a smoother surface compared to that of the sodium alginate film. This was due to the fact that *Bacillus subtilis* worked well as a reinforcing material in the reinforcement and development of the microbial composite films, as bacteria were able to fill into the voids and close up the gaps between the components in the film. The higher the mass of bacteria, the bigger the surface area for interaction [20]. Thus, more protein–alginate interaction and bonding can occur, which results in higher tensile strength. Therefore, 0.5 g microbial composite films that were reinforced with *Bacillus subtilis* had a smoother surface and better overall quality than sodium alginate films. Figure 5 shows cross-sectional scanning electron microscope (SEM) images of sodium alginate film and 0.5 g microbial composite film. Figure 6 shows surface scanning electron microscope (SEM) images of sodium alginate film and 0.5 g microbial composite film. 

### 3.5. Optimum Conditions of Microbial Composite Film for Physical and Mechanical Properties 

Based on the results obtained, it was found that reinforcing sodium alginate films with *Bacillus subtilis* cell mass, overall, provided reinforcing effects to the films, especially as the bacterial cell mass increased. Among all the bacterial cell masses utilized, it was shown that 0.5 g microbial composite film had the best effect of reinforcement on the sodium alginate films. The optimal conditions of the microbial composite films in physical and mechanical analyses as mentioned above are shown in Table 3. Therefore, additional analyses were carried out in order to characterize the microbial composite films further.

### 3.6. Chemical Analysis 

The chemical analysis checked the chemical properties of the microbial composite films. This was performed so that more details regarding the chemical properties could be obtained for an in-depth study about the microbial composite films. The chemical analysis included in this research project was Fourier transform infrared spectrometry (FTIR). 

### 3.7. Fourier Transform Infrared Spectrometry (FTIR)

In this research, attenuated total reflectance-Fourier transform infrared (ATR-FTIR) was used, as the samples of this research project were in the form of films. When comparing the two figures, there was no huge difference between the transmittance of sodium alginate film and 0.5 g microbial composite film. Transmittance at a certain wavenumber of FTIR is able to provide information about the chemical bond that is present in the material, in this case, the sodium alginate film and microbial composite film. The chemical bonds that could be found in both sodium alginate film and 0.5 g microbial composite films were OH stretching at about 3327 cm^−1^, CH symmetrical stretching at about 2883 cm^−1^, OH bending of absorbed water at about 1623 cm^−1^, HCH and OCH in-plane bending vibration at about 1423 cm^−1^, CH_2_ rocking vibration at C6 at about 1314 cm^−1^, C–C, C–OH, C–H ring and side group vibrations at about 1046, 1020, 994 and 895 cm^−1^. The only differences of sodium alginate film and 0.5 g microbial composite film were that sodium alginate film lacked values at 800 and 662 cm^−1^ that referred to C–C, C–OH, C–H ring and side group vibrations and C–OH out-of-plane bending, respectively. This indicated that microbial composite film possessed more chemical functional groups that contributed to their physical, mechanical and chemical properties than sodium alginate films. In this case, hydrogen bonding occurred between the hydroxide groups of sodium alginate components and the protein molecules presented on the outer membrane of bacteria cells [14]. The hydroxide groups of sodium alginate were able to form hydrogen bonds with protein molecules [21,22]. The protein–alginate hydrogen bonding that occurred in the composite film adjoined the bacteria and the components in the film, which strengthened the inner area of the film. Figure 7 shows Fourier transform infrared spectroscopy (FTIR) graph of sodium alginate film and 0.5 g microbial composite film. Table 4 shows the conductivity and Fourier transform infrared spectrometry (FTIR) of sodium alginate film and microbial composite films.

## 4. Thermal Analysis 

### 4.1. Differential Scanning Calorimetry (DSC)

Differential scanning calorimetry is a thermoanalytical technique in which, as a function of temperature, the difference in the amount of heat needed to increase the temperature of the sample and reference is determined. In this research project, the melting points of the films were determined with an initial temperature of 30 °C and an ending temperature of 400 °C, where an increasing temperature of 10 °C/min was used. The sample was analyzed in an inert gas atmosphere of nitrogen in the first run of heating. Referring to Figure 8, it could be seen that the graphs of sodium alginate films without bacterial cells and 0.5 g microbial composite films were almost the same. The first downward loops or endothermic band correspond to the evaporation of hydration water molecules from the films. The emergence of a sharp endothermic band at 210 °C probably corresponded to cleavage enthalpies such as breakage of bonds within the complex. Such sharp endothermic band indicates a highly ordered (crystallite) molecular arrangement forming the so-called “egg-box” structure within the calcium alginate in the microbial composite film [23]. This sudden decline in the graph could be referred to the melting point of the microbial composite films. The melting point of the sodium alginate film was 212.56 °C, while the melting point of the 0.5 g microbial composite film was 218.94 °C. It is shown that the 0.5 g microbial composite film had a higher melting point than the sodium alginate film. This was due to the fact that the bacterial cell mass provided a reinforcing effect to the microbial composite film, as it is proven that bacterial cells or the cellulose contained inside can reinforce composite materials and improve their mechanical properties [24]. This also means that the bacterial cell mass successfully reinforced the composite films. Figure 8 shows differential scanning calorimetry (DSC) graphs of sodium alginate film and 0.5 g microbial composite film. 

### 4.2. Thermogravimetric Analysis (TGA)

Thermogravimetric analysis (TGA) is an analytical technique used to determine the thermal stability of a material was used to determine the decomposition temperature of the films in this research. Heating of 10 °C/min was used to heat up the microbial composite films. The sample was analyzed in an inert gas atmosphere of nitrogen. Referring to Figure 9, the first downward sloping curve of the graph shows the evaporation of water content from the films. The descending TGA thermal curve indicates that weight loss occurred. The steepest slope observed around 250 °C refers to the decomposition temperature of the films. It could be deduced that the TGA graph for 0.5 g microbial composite film had a higher onset decomposition temperature compared to that of the sodium alginate film. It was found that in the 0.5 g microbial composite film, the onset decomposition temperature was 252.69 °C, while in the sodium alginate film it was 248.02 °C. The mass loss of soot due to combustion or pyrolysis in the temperature range from 400 to 800 °C can be clearly seen. As a result of the mass loss, the mass remaining at 800 °C was substantially reduced, reducing the mass of soot in the sample [25]. Sodium alginate film that was crosslinked by calcium chloride released the calcium ions into the alginate solution during the crosslinking process. It contained a number of high-temperature components that required energy to dissociate the tightly crosslinked calcium alginate [26]. Therefore, the onset decomposition temperature for the sodium alginate film was 248.02 °C, although this number was slightly lower compared to those of the microbial composite films. The slight difference between the decomposition temperature of sodium alginate film and the 0.5 g microbial composite films could be due to the fact that *Bacillus subtilis* cell masses that were introduced into the sodium alginate strengthened the bonds between the bacteria and sodium alginate and thus enhanced the thermal stability of the microbial composite films. Figure 9 shows the thermogravimetric analysis (TGA) graphs of sodium alginate film and 0.5 g microbial composite film. Table 5 (in the Appendix A) shows differential scanning calorimetry (DSC) and thermogravimetric analysis (TGA) of sodium alginate and microbial composite films. 

## 5. Conclusions

In conclusion, it was shown that films reinforced with *Bacillus subtilis* had all the properties mentioned significantly improved. Microbial composite film reinforced with 0.5 g of *Bacillus subtilis* had the highest tensile strength, breaking strain and toughness, which were 0.858 MPa, 87.406% and 0.045 MJ/m^3^, respectively. The thickness of the 0.5 g film was 1.057 mm, while white light opacity, black light opacity and brightness values were 13.65%, 40.55% and 8.19%, respectively. It also had the highest conductivity, which was 37 mV. Its water absorption ability was 300.93%. Furthermore, it had a high melting point of 218.94 °C and high decomposition temperature of 252.69 °C. SEM also showed that it had a filled cross-sectional structure and a smoother surface compared to that of the sodium alginate film. Additionally, FTIR showed that 0.5 g microbial composite films possessed more functional groups at 800 and 662 cm^−1^ wavenumbers that referred to C–C, C–OH, C–H ring and side group vibrations and C–OH out-of-plane bending, respectively, which contributed to the stronger bonds in the microbial composite film. Therefore, overall, sodium alginate films reinforced with bacterial cell mass had better properties than the sodium alginate film. It was shown that *Bacillus subtilis* has potential to be used as a reinforcing material in the development of microbial composite films and the prospect to be used in electronic applications, because the conductivity of the films increased with increasing *Bacillus subtilis* cell mass. Table 6 summarizes the comparison between the properties of sodium alginate film (control) and 0.5 g microbial composite film.

## Figures and Tables

**Figure 1 polymers-13-02103-f001:**
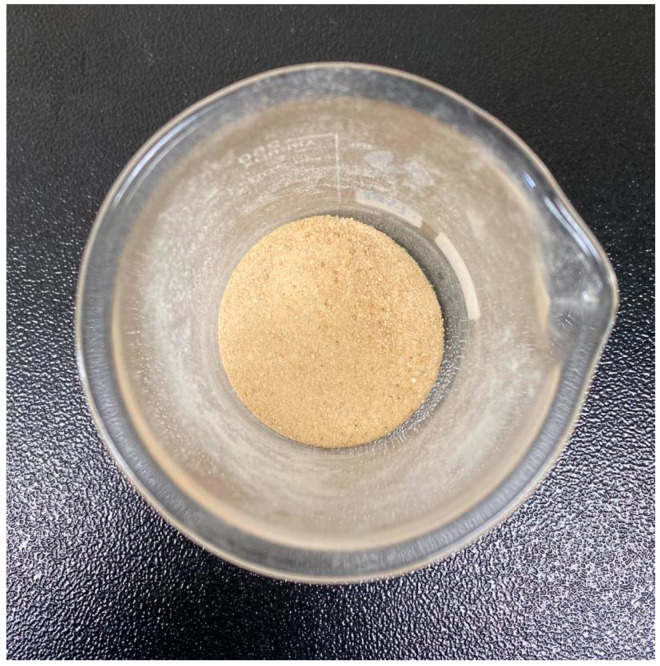
Freeze-dried *Bacillus subtilis*.

**Figure 2 polymers-13-02103-f002:**
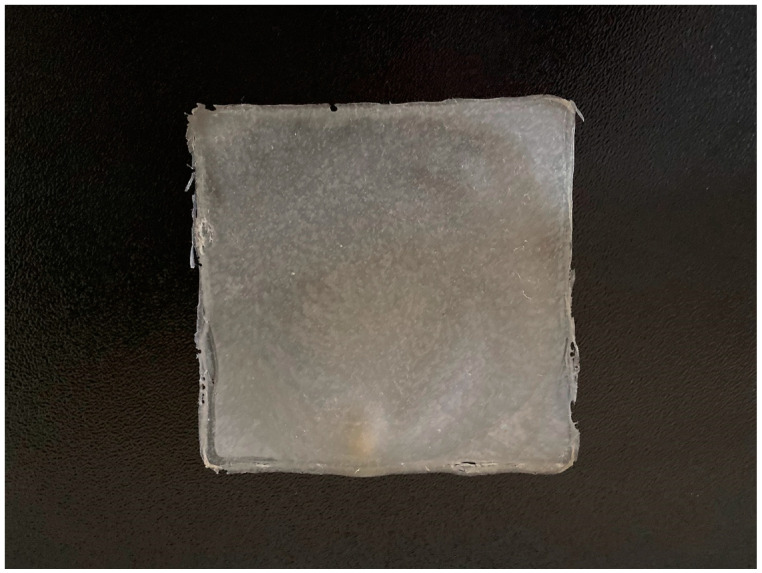
Microbial composite film, dimensions 16 cm (length) × 16 cm (width).

**Figure 3 polymers-13-02103-f003:**
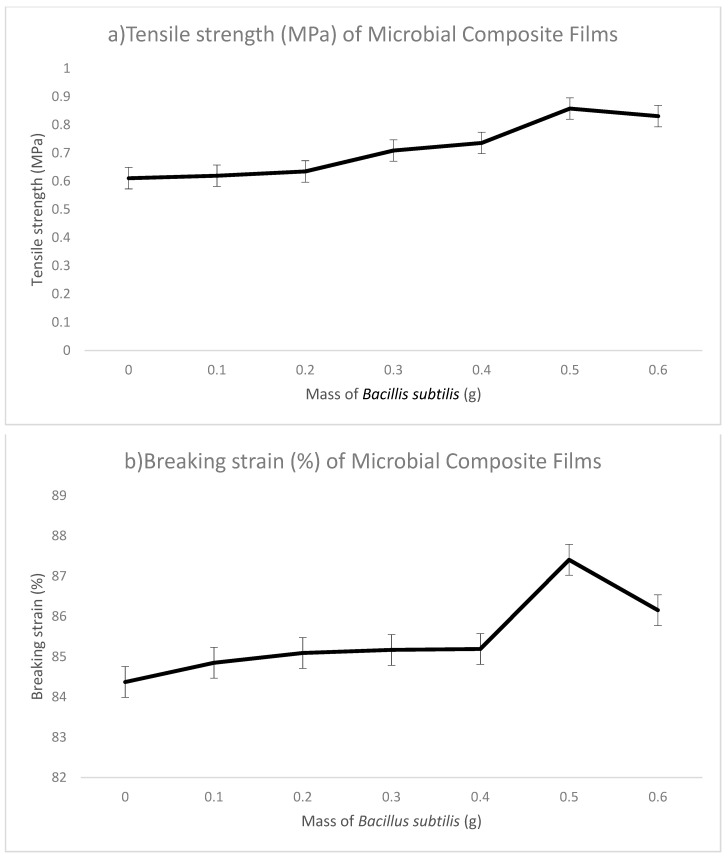
Mechanical properties of microbial composite films with various mass of *Bacillus subtilis*. (**a**) Tensile strength, (**b**) breaking strain and (**c**) toughness. (*N* = 3).

**Figure 4 polymers-13-02103-f004:**
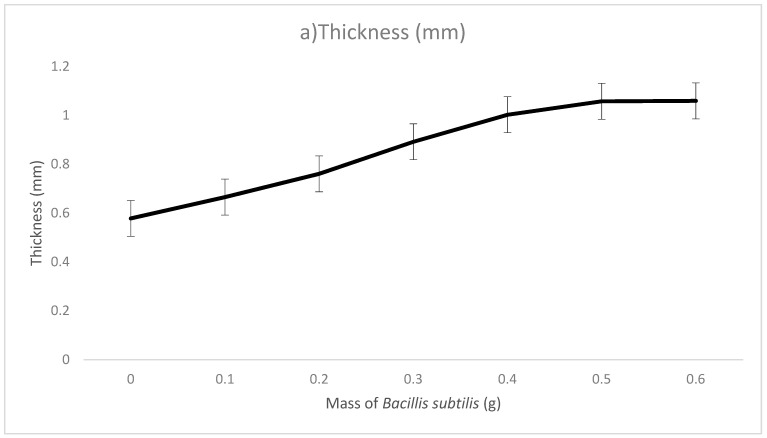
Graphs of (**a**) thickness, (**b**) opacity (white light), (**c**) opacity (black light), (**d**) brightness, (**e**) conductivity and (**f**) water absorption of microbial composite films. (*N* = 3).

**Figure 5 polymers-13-02103-f005:**
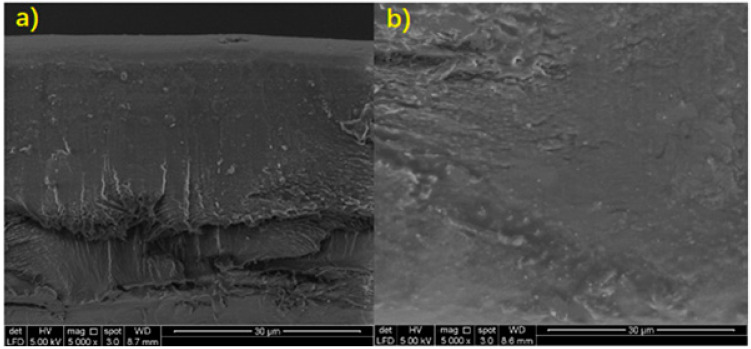
Cross-sectional scanning electron microscope (SEM) image of (**a**) sodium alginate film and (**b**) 0.5 g microbial composite film.

**Figure 6 polymers-13-02103-f006:**
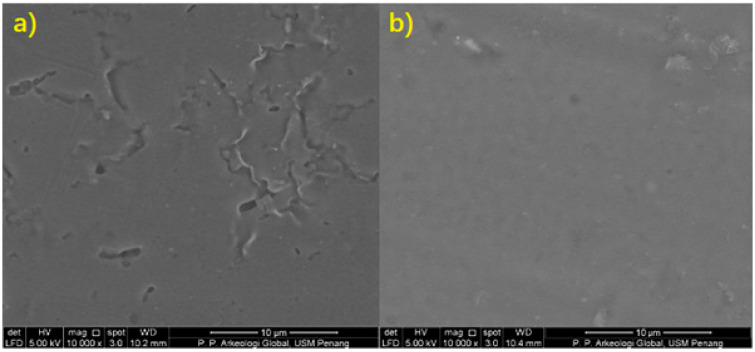
Surface scanning electron microscope (SEM) image of (**a**) sodium alginate film and (**b**) 0.5 g microbial composite film.

**Figure 7 polymers-13-02103-f007:**
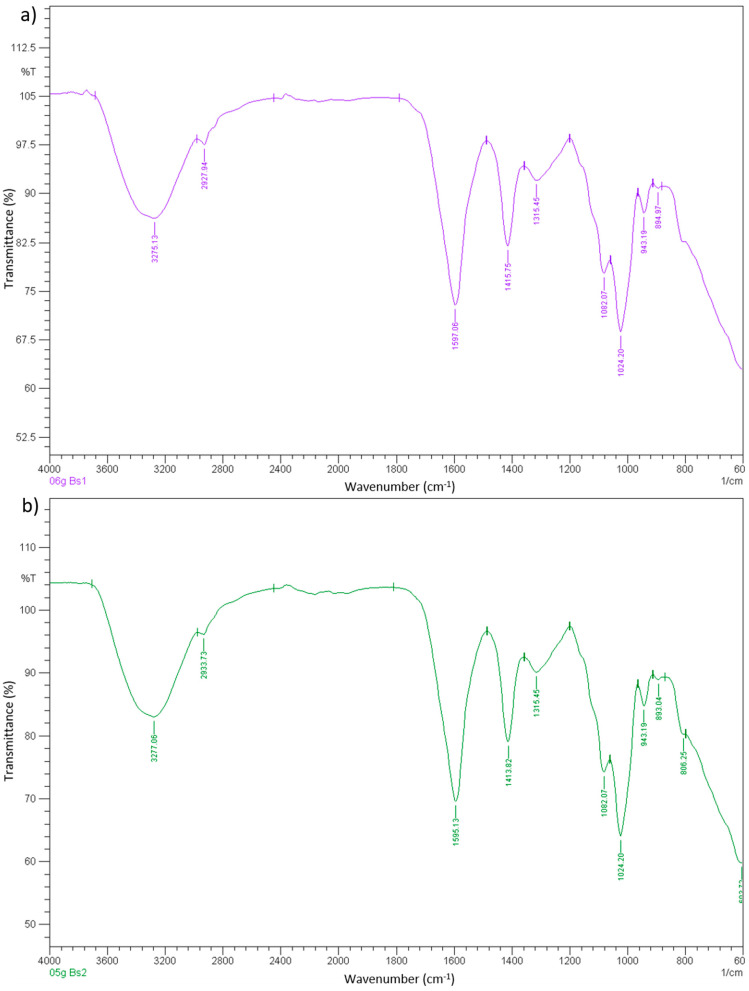
Fourier transform infrared spectroscopy (FTIR) graph of (**a**) sodium alginate film and (**b**) 0.5 g microbial composite film.

**Figure 8 polymers-13-02103-f008:**
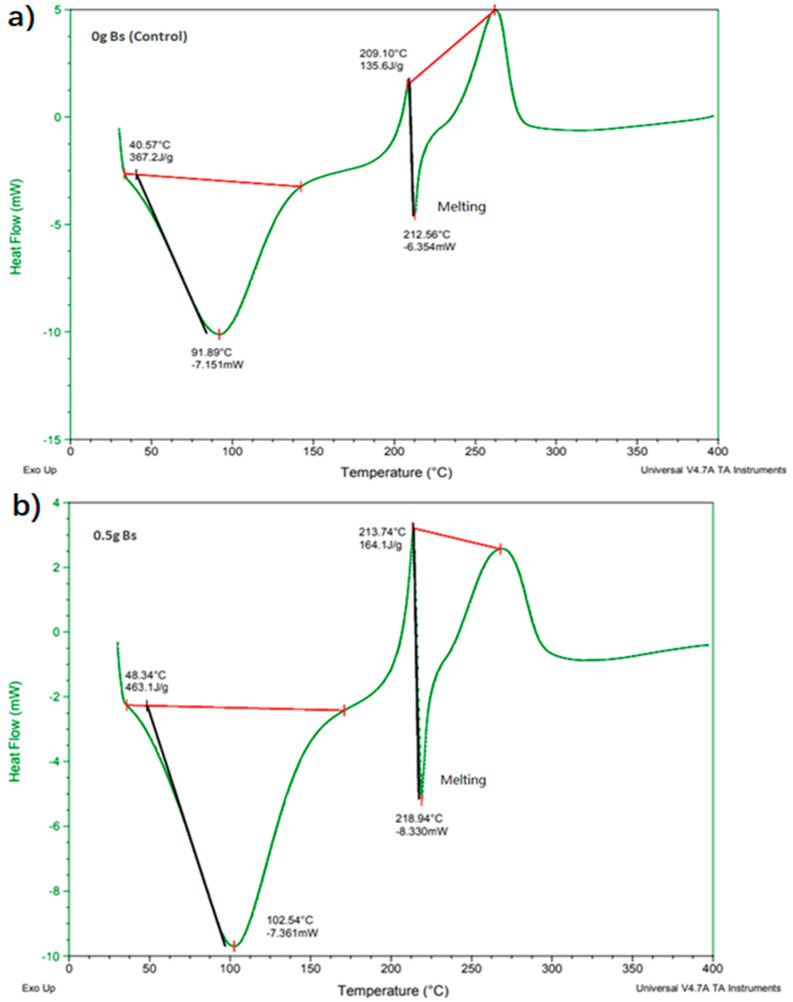
Differential scanning calorimetry (DSC) graph of (**a**) sodium alginate film and (**b**) 0.5 g microbial composite film.

**Figure 9 polymers-13-02103-f009:**
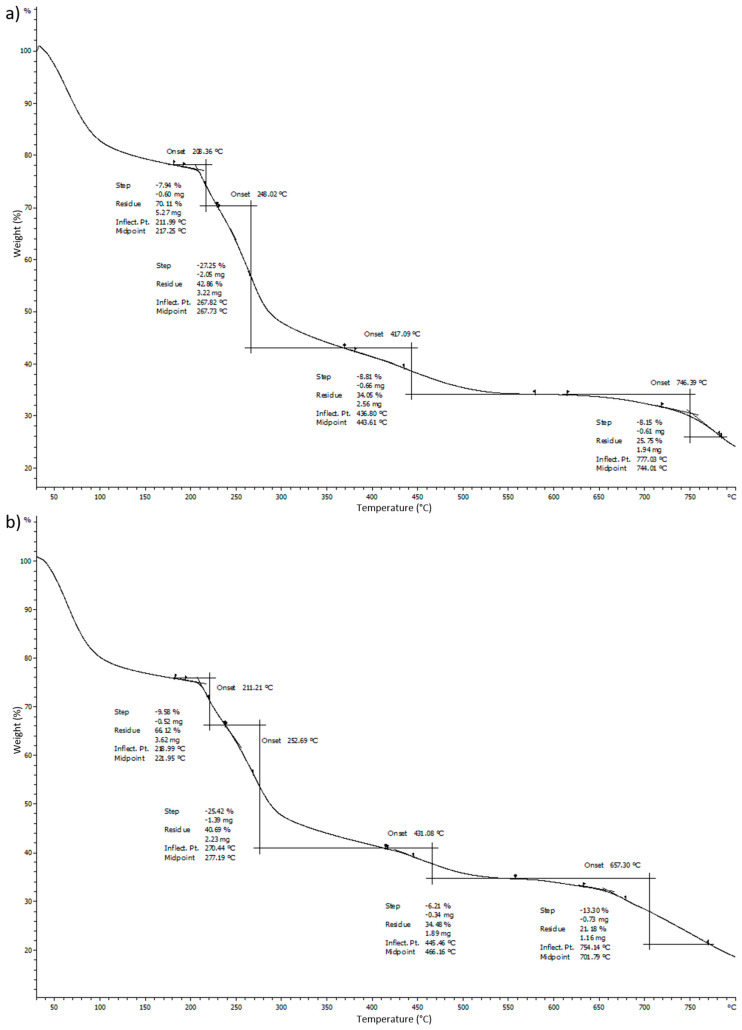
Thermogravimetric analysis (TGA) graph of (**a**) sodium alginate film) and (**b**) 0.5 g microbial composite film.

**Table 1 polymers-13-02103-t001:** Tensile strength (MPa), breaking strain (%) and toughness (MJ/m^3^) of sodium alginate film and microbial composite films.

	Mass of *Bacillus subtilis* (g)	Tensile Strength (MPa)	Breaking Strain (%)	Toughness (MJ/m^3^)
**Sodium Alginate Film**	**0**	0.611	84.372	0.016
**Microbial Composite Films**	**0.1**	0.620	84.849	0.016
	**0.2**	0.635	85.092	0.018
	**0.3**	0.709	85.169	0.024
	**0.4**	0.736	85.192	0.024
	**0.5**	0.858	87.406	0.045
	**0.6**	0.831	86.155	0.310
**Conclusions**		When mass of *Bacillus subtilis* increased, tensile strength increased.	When mass of *Bacillus subtilis* increased, breaking strain increased.	When mass of *Bacillus subtilis* increased, toughness increased.
**Remarks**		The highest tensile strength recorded in 0.5 g.	The highest breaking strain recorded in 0.5 g.	The highest toughness recorded in 0.5 g.

**Table 2 polymers-13-02103-t002:** Thickness (mm), opacity (white light and black light, %), brightness (%), conductivity and water absorption (%) and scanning electron microscopy (SEM) of sodium alginate film and microbial composite films.

	Mass of *Bacillus subtilis* (g)	Thickness (mm)	Opacity (%)		Brightness (%)	Conductivity		Water Absorption (%)	Scanning Electron Microscopy (SEM)
			White Light	Black Light		mV	S/m		
**Sodium Alginate Film**	**0**	0.578	7.52	38.41	4.24	11.33	0.74	264.29	Rough surface Voids on cross-sectional diagram
**Microbial Composite Films**	**0.1**	0.665	8.1	37.86	6.65	12	0.24	269.05	Smooth surface Filled voids on cross-sectional diagram
	**0.2**	0.761	8.39	38.61	6.67	16.33	0.33	270.83	
	**0.3**	0.892	9.19	39.29	7.38	17.33	0.35	279.17	
	**0.4**	1.002	12.16	41.69	7.88	19.67	0.39	289.35	
	**0.5**	1.057	13.65	40.55	8.19	37	0.74	300.93	
	**0.6**	1.059	16.45	41.18	9.3	26.33	0.53	307.78	
**Conclusions**		When mass of *Bacillus subtilis* increased, thickness increased.	When mass of *Bacillus subtilis* increased, opacity increased.		When mass of *Bacillus subtilis* increased, brightness increased.	When mass of *Bacillus subtilis* increased, conductivity increased.		When mass of *Bacillus subtilis* increased, water absorption increased.	When mass of *Bacillus subtilis* increased, conductivity increased.
**Remarks**		The highest thickness recorded with 0.6 g.	The highest opacity recorded with 0.6 g.		The highest brightness recorded with 0.6 g.	The highest conductivity recorded with 0.5 g.		The highest water absorption recorded with 0.6 g.	The highest conductivity recorded with 0.5 g.

**Table 3 polymers-13-02103-t003:** Optimal conditions of 0.5 g microbial composite film for physical and mechanical properties.

Physical and Mechanical Analysis	Value
**(1) Tensile strength (MPa)**	0.858
**(2) Breaking strain (%)**	87.406
**(3) Toughness (MJ/m^3^)**	0.045
**(4) Thickness (mm)**	1.057
**(5) Opacity (white light, %)**	13.650
**(6) Opacity (black light, %)**	40.550
**(7) Brightness (%)**	8.190
**(8) Conductivity (mV)**	37.000
**(9) Water absorption (%)**	300.930
**(10) Scanning electron microscopy (SEM)**	Microbial composite film has smoother surface and filled voids.

**Table 4 polymers-13-02103-t004:** Conductivity and Fourier transform infrared spectrometry (FTIR) of sodium alginate film and microbial composite films.

	Mass of *Bacillus subtilis* (g)	Fourier Transform Infrared Spectrometry (FTIR)
**Sodium alginate film**	**0**	Bonds: OH stretching, C–H symmetrical stretching, OH bending of absorbed water, HCH and OCH in-plane bending vibration, CH_2_ rocking vibration at C6, C–C, C–OH, C–H ring and side group vibrations, COC, CCO and CCH deformation and stretching.
**Microbial composite films**	**0.1**	Bonds: OH stretching, C–H symmetrical stretching, OH bending of absorbed water, HCH and OCH in-plane bending vibration, CH_2_ rocking vibration at C6, C–C, C–OH, C–H ring and side group vibrations, COC, CCO and CCH deformation and stretching, C–OH out-of-plane bending.
	**0.2**	
	**03**	
	**0.4**	
	**0.5**	
	**0.6**	
**Conclusions**		Microbial composite films possessed more chemical bonds compared to sodium alginate films, which contributed to stronger properties.
**Remarks**		Microbial composite films had extra wavenumbers at ~800 and ~662cm^−1^ compared to sodium alginate films.

**Table 5 polymers-13-02103-t005:** Differential scanning calorimetry (DSC) and thermogravimetric analysis (TGA) of sodium alginate and microbial composite films.

	Mass of *Bacillus subtilis* (g)	Differential Scanning Calorimetry (DSC)	Thermogravimetric Analysis (TGA)
**Sodium alginate film**	**0**	212.56	248.02
**Microbial composite films**	**0.1**	216.01	249.88
	**0.2**	216.55	249.64
	**03**	217.69	250.72
	**0.4**	217.87	251.12
	**0.5**	218.94	252.69
	**0.6**	218.58	252.52
**Conclusions**		When mass of *Bacillus subtilis* increased, melting point increased.	When mass of *Bacillus subtilis* increased, decomposition temperature increased.
**Remarks**		The highest melting point recorded with 0.5 g.	The highest decomposition temperature recorded with 0.6 g.

**Table 6 polymers-13-02103-t006:** Comparison between the properties of sodium alginate film (control) and 0.5 g microbial composite film.

Physical, mechanical and chemical Analysis	Sodium Alginate Film (Control)	0.5 g Microbial Composite Film
**(1) Tensile strength (MPa)**	0.611	0.858
**(2) Breaking strain (%)**	84.372	87.406
**(3) Toughness (MJ/m^3^)**	0.016	0.045
**(4) Thickness (mm)**	0.578	1.057
**(5) Opacity (white light, %)**	7.520	13.650
**(6) Opacity (black light, %)**	38.410	40.550
**(7) Brightness (%)**	4.240	8.190
**(8) Conductivity (mV)**	11.220	37.000
**(9) Water absorption (%)**	264.290	300.930
**(10) Differential scanning calorimetry, DSC (°C)**	212.560	218.940
**(11) Thermogravimetric analysis, TGA (°C)**	248.020	252.690
**(12) Scanning electron microscopy (SEM)**	Rougher cross-sectional and surface	Filled cross-sectional and smoother surface
(13) Fourier transform infrared spectrometry (FTIR)	Lack of values at 800 and 662 cm^−1^ wavenumbers.	Extra values at 800 and 662 cm^−1^ wavenumbers for C–C, C–OH, C–H ring and side group vibrations and C–OH out-of-plane bending.

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
