# Peer review of "Elucidation of Mechanical, Physical, Chemical and Thermal Properties of Microbial Composite Films by Integrating Sodium Alginate with Bacillus subtilis sp."

_polymers, 2021, doi:10.3390/polym13132103_

Round 1
Reviewer 1 Report
This research investigates the mechanical, physical, chemical, and thermal properties of the microbial composite films by integrating sodium alginate with Bacillus subtilis. It shows that the composite films could significantly improve all these above properties. I think some explanation in this paper may not be adequate and suggest following major revisions.
- Introduction part. Try to make this part more concise, especially the first two paragraphs. In addition, the authors should briefly mention other researchers’ works on developing microbial composite films.
- I find the units used in this paper are quite confusing. For example, the authors employ 0.1g, 0.2g and so on to classify different tests. These values are meaningless to readers if different amount of sodium alginate is used. Why not use the ratio of Bacillus subtilis to sodium alginate instead? Similarly, the authors use mV for conductivity. Why not use the SI units (S/m)?
- Page 7 Figure 3. What does the “N=3” mean in the caption; three tests for each case? If so, are the results statistically significant, providing so few tests were done?
- The authors should revise all the tables and figures in this paper. For example, Figure 3 contains all the information in Table 1, thus Table 1 is redundant. Moreover, try to put the two panels in figure 7, 8, or 9 together, in order to better show the differences.
- Why does the 0.6 g microbial composite film show different trend of mechanical properties (lower tensile strength, breaking strain, and toughness than 0.5g)? More explanation should be given.
- Page 19. The authors mention that the degradation temperature for composite film is 252.69 degree and the sodium alginate film is 248.02 degree. The technical details about how to obtain these values from Figure 9 should be shown in Supplemental Information. In addition, the words in Figure 7, and 9 are too small to be seen.
Author Response
Good day Reviewers, please kindly find the attachment for your perusal. Thanks!

Reviewer 2 Report
Authors prepared sodium alginate films containing various amounts of Bacillus subtilis and characterized the obtained materials by various methods. While the subject could be interesting, there is a major drawback coming from improper selection of material amount when preparing samples. Since various amounts of Bacillus subtilis were added to same amounts of sodium alginate and films were formed in shapes of same surface, the thickness of the films varies, affecting all the measured properties. Correct approach would be to change the ratio of components while maintaining constant the total sample mass, thus similar thickness of obtained films. Only then discussion on properties could be properly corelated with sample composition.
Author Response

(The authors gave the same response as above.)

Reviewer 3 Report
The manuscrit by Wai Chun et al . (Elucidation of Mechanical, Physical, Chemical and Thermal Properties of Microbial Composite Films by Integrating Sodium Alginate with Bacillus subtilis sp.) reports the development of microbial composite films by integrating sodium alginate with Bacillus subtilis. They investigated the mechanical, physical, chemical and thermal properties of microbial composite films in detail. The detected that the amount of the microbial content has significant effect on the all properties.
In general, the report is well-written and easy-to-follow. I think the manuscript provides valuable in formation to this research field. I recommend the publication of the report, after some minor point given below are addressed.
The abstract seems to be short and incomplete. I think it would be better to add some major results in the Abstract section.
The first two paragraphs of the Introduction section should be shortened. The authors must provide some more Refererences on fabrication and applications of microbial composite films.
The authors should provide the presence of bacteria inside the thin film through the detailed SEM and confocal microscopy images (similar to work by Zajdel et al. October 2018 Scientific Reports 8(1) DOI: 10.1038/s41598-018-33521-9 ). These data would provide more insight to discuss the all the report.
It is not easy to read the data on Figure 9. Please improve the quality of the image.
Author Response

(The authors gave the same response as above.)

Round 2
Reviewer 1 Report
Accept in present form
Author Response
Thank you for your valuable time!
Reviewer 2 Report
- Mechanical properties, opacity and water absorption strongly depend on film thickness, which varies for studied samples due to selected preparation procedure. Therefore discussions on these properties are meaningless, since the effect of microbial component cannot be individually observed.
- Interpretation of DSC results is totally wrong. The sudden decrease of heat flow around 210 oC is an artifact, most probably due to a burst mass loss from the DSC crucible. One cannot consider melting around 210 oC since TG curves in Fig 9 shows continuous mass loss from sample. Gas atmosphere (inert or oxidative) was not mentioned. Also, it was not mentioned if data were recorded for the first or the second heating run.
- TGA data is poorly discussed. It was not mentioned if TG was performed in inert or oxidative atmosphere. The Tg curves as well as the related DTG curves should be overlapped for all samples for clear visualization of effects induced by the addition of microbial component. The slopes from 400 to 800 oC cannot be related neither to residual mass, nor to ash content. It is not clear what the temperatures in Table 5 stands for, and why so great variation for TG.
Author Response
Thank you for your valuable time!
Please kindly find the attachment for the second round of responses based on the comments made.
Thanks again and stay safe!

Reviewer 3 Report
The changes performed by the authors improved the quality of the report. I recommend of the manuscript in its present form.
Author Response
Thank you for your valuable time!